# Hope and Hype around Immunotherapy in Triple-Negative Breast Cancer

**DOI:** 10.3390/cancers15112933

**Published:** 2023-05-26

**Authors:** Flavia Jacobs, Elisa Agostinetto, Chiara Miggiano, Rita De Sanctis, Alberto Zambelli, Armando Santoro

**Affiliations:** 1Department of Biomedical Sciences, Humanitas University, Via Rita Levi Montalcini 4, 20090 Pieve Emanuele, MI, Italy; flavia.jacobs@humanitas.it (F.J.); chiara.miggiano@humanitas.it (C.M.); rita.de_sanctis@hunimed.eu (R.D.S.); alberto.zambelli@hunimed.eu (A.Z.); 2Academic Trials Promoting Team, Institut Jules Bordet, L’Université Libre de Bruxelles (U.L.B.), 1070 Brussels, Belgium; elisa.agostinetto@hubruxelles.be; 3IRCCS Humanitas Research Hospital, Humanitas Cancer Center, Via Manzoni 56, 20090 Pieve Emanuele, MI, Italy

**Keywords:** triple-negative breast cancer, immunotherapy, immune checkpoint inhibitors (ICIs), predictive biomarkers, immune-related toxicity

## Abstract

**Simple Summary:**

Triple-negative breast cancer (TNBC) is the most aggressive breast cancer subtype accounting for approximately 10–20% of all cases. Due to a lack of actionable targets, chemotherapy has been for many decades the preferred, and often the only, available treatment option for this disease. There is now evidence, from several randomized controlled trials, that immune checkpoint inhibitors are effective as first-line treatment for advanced TNBC expressing PD-L1 and as neoadjuvant chemotherapy for high-risk early TNBC. Despite these encouraging results, there are still many open issues about the optimal use of immunotherapy in TNBC. This review summarizes the main results from clinical trials testing immunotherapy in TNBC and critically discusses some limitations of these study results. Finally, we present the challenges that need to be addressed soon in the evolving field of immune-oncology.

**Abstract:**

Triple-negative breast cancer (TNBC) holds a poor prognosis compared to other breast cancer subtypes, and the development of new effective treatment strategies is an unmet medical need. TNBC has traditionally been considered not amenable to treatment with targeted agents due to a lack of actionable targets. Therefore, chemotherapy has remained the mainstay of systemic treatment for many decades. The advent of immunotherapy raised very hopeful expectations in TNBC, possibly due to higher levels of tumor-infiltrating lymphocytes, PD-L1 expression and tumor mutational burden compared to other breast cancer subtypes, that predict an effective anti-tumor immune-engagement. The results of clinical trials testing immunotherapy in TNBC led to the approval of the combination of immune checkpoint inhibitors and chemotherapy in both early and advanced settings. However, some open questions about the use of immunotherapy in TNBC still exist. These include a deeper understanding of the heterogeneity of the disease, identification of reliable predictive biomarkers of response, determination of the most appropriate chemotherapy backbone and appropriate management of potential long-term immune-related adverse events. In this review we aim to examine the available evidence on the use of immunotherapy strategies in both early and advanced TNBC, to critically discuss some of the limitations encountered in clinical research and to summarize data on novel promising immunotherapeutic strategies beyond PD-(L)1 blockade that have been investigated in the most recent trials.

## 1. Introduction

Triple-negative breast cancer (TNBC) accounts for 10 to 20% of newly diagnosed invasive breast cancers and is defined by the absence of immunostaining for estrogen receptor (ER), progesterone receptor (PgR) and human epidermal growth factor receptor-2 (HER2) [1]. Among breast cancer subtypes, TNBC exhibits the worst clinical outcome, with more than 30% of patients developing metastatic disease and relapses during the first 2–3 years from diagnosis [2]. Metastatic TNBC has a poor prognosis with a median overall survival (OS) of approximately 17.5 months [3]. This poor prognosis reflects an intrinsic aggressive behavior, as TNBC tends to have unfavorable features such as a high proliferative rate (Ki67), high tumor grade and an invasive phenotype [4]. Moreover, the lack of actionable oncogenic targets was the reason why chemotherapy represented the only valid therapeutic option until a few years ago [5]. 

A deeper understanding of the molecular characteristics of TNBC unveiled different subtypes so that TNBC is no longer considered as a single entity [6,7]. Based on gene expression profiles, in 2011 Lehmann et al. identified six different TNBC molecular subtypes: (I) basal-like 1 (BL1), (II) basal-like 2 (BL2), (III) immunomodulatory (IM), (IV) mesenchymal (MES), (V) mesenchymal stem-like (MSL), (VI) luminal androgen receptor (LAR) [8]. Subsequently, Burstein et al. proposed a simplified molecular classification with four different subtypes including LAR, MES, basal-like immunosuppressed (BLIS) and basal-like immune activated (BLIA) [9]. Each molecular subtype shows different mutational profiles and signaling pathways [10]. Nonetheless, these classifications have had no direct implications in clinical practice, so far. 

Immunotherapy has emerged as a significant advancement in oncology, but it took years before immune checkpoint inhibitors (ICIs) were investigated in TNBC. The discovery of higher genomic instability and a better understanding of the importance of tumour-infiltrating lymphocytes (TILs), programmed death ligand 1 (PD-L1) expression and tumor microenvironment (TME), paved the way for clinical trials testing ICIs in TNBC [11]. The results of these trials led to the approval of ICIs in combination with chemotherapy for both early and advanced TNBC. However, challenges still exist in integrating immunotherapy into TNBC treatment algorithms, and several questions remain unanswered [12,13,14,15].

In this review, we aim to explore the available evidence supporting the use of immunotherapy in the treatment of TNBC in both early and advanced settings, discuss the limitations of the studies conducted so far, and critically analyze the open questions that require further investigation. Additionally, we provide insights into the most recent trials that are investigating the combination of immunotherapies with ICIs and conventional therapies, as well as other immunotherapeutic strategies.

## 2. Main Studies with Immunotherapy

### 2.1. Advanced Triple-Negative Breast Cancer

#### 2.1.1. Immune-Checkpoint Inhibitors in Monotherapy

Primary investigations conducted with immunotherapy in TNBC evaluated the safety and efficacy of this strategy as monotherapy for patients with advanced disease. (Table 1).

The phase Ib KEYNOTE-012 trial [16] was the first study to investigate the use of pembrolizumab in patients with advanced PD-L1-positive TNBC, gastric cancer, urothelial cancer, and head and neck cancer. Among 27 evaluable TNBC women, pembrolizumab showed promising efficacy with an objective response rate (ORR) of 18.5% and a disease control rate (DCR) of 37.5%. Subsequently, the phase II KEYNOTE-086 trial [17,18] evaluated the efficacy of pembrolizumab either in previously treated (cohort A) or untreated (cohort B) metastatic TNBC. In cohort A, where half of the patients were pretreated with more than 3 lines of therapy, pembrolizumab monotherapy showed marginal benefit with an ORR of only 5% [17]. However, in cohort B of the study, first-line pembrolizumab induced an ORR of 21% [18]. These two studies also showed a higher rate of responses and greater efficacy when pembrolizumab was used in PD-L1 positive patients, in those with lower tumor mutational burden (TMB) and in the absence of visceral disease. However, very few long-lasting responses were observed.

Unfortunately, the following phase III trial, the KEYNOTE-119 [19], failed to demonstrate superiority of pembrolizumab monotherapy versus chemotherapy beyond the first line in metastatic TNBC. Although no differences were detected in OS, there was a positive trend towards benefit for patients with combined positive score (CPS ≥ 10) treated with pembrolizumab.

The positive results of KEYNOTE-158 [20,21] led to the histology-agnostic FDA approval in June 2020 of pembrolizumab for the treatment of patients with unresectable or metastatic solid tumors with high TMB (defined as ≥10 mutations/megabase determined by an FDA-approved test), who have no alternative treatment options and have progressed following prior treatments. It is essential to consider that the clinical trial data analyzed for this histology-agnostic approval did not include breast cancer patients. Nevertheless, a subsequent study, the TAPUR trial, showed that pembrolizumab monotherapy had a good DCR (37%) and ORR (21%) in 28 patients with previously treated metastatic breast cancer with high TMB, supporting the use of pembrolizumab also in this subgroup of breast cancer patients [22]. 

Several clinical trials have tested the efficacy of other ICIs in monotherapy for the treatment of advanced TNBC. These include the phase I JAVELIN trial with avelumab [23], the phase II ENCORE602/TRIO025 trial with atezolizumab [24], and the phase II TONIC trial with nivolumab [25]. The SAFIRO2-BREAST IMMUNO trial evaluated the use of durvalumab as a maintenance therapy for HER2-negative breast cancer patients who had undergone 6–8 cycles of chemotherapy without experiencing disease progression. While the trial’s results did not show an improvement in progression-free survival (PFS) or OS in the overall patient population, in a subgroup analysis of 82 TNBC patients, durvalumab significantly improved OS. Specifically, the median OS in the durvalumab group was 21.2 months, compared to 14.0 months in the maintenance chemotherapy group (HR 0.54, *p* = 0.0377). Additionally, an exploratory analysis of TNBC patients with PD-L1-positive disease (*n* = 32) revealed a lower hazard ratio (HR) of death compared to those with PD-L1-negative TNBC (*n* = 29) [26]. The results of these trials are reported in Table 1.

**Table 1 cancers-15-02933-t001:** Main studies with immune-checkpoint inhibitors in monotherapy for the treatment of advanced triple-negative breast cancer.

Study Name	Phase	Line of Therapy and Population	TNBC (=N)	Treatment	Findings
**KEYNOTE-012** [16]	Ib	mTNBC All lines (first in 15.6%)	32	Pembrolizumab	ORR: 18.5% mPFS: 1.9 mo OS: 11.2 mo
**JAVELIN** [23]	I	mTNBC All lines (first or second line in 50% of patients)	58	Avelumab	ORR (TNBC cohort): 5.2% ORR (PD-L1 positive): 22.2% ORR (PD-L1 negative): 2.6%
**KEYNOTE-086** **(cohort A)** [17]	II	mTNBC Second or later lines (second in 31%)	170	Pembrolizumab	ORR (ITT): 5.3% ORR (PD-L1 positive): 5.7% ORR (PD-L1 negative): 4.7%
**KEYNOTE-086** **(cohort B)** [18]	II	mTNBC First line	84	Pembrolizumab	ORR: 21.4% mPFS: 2.1 mom OS: 18 mo
**TAPUR** [22]	II	mBC with HTMB Second or later lines (≥3 in 93%)	28	Pembrolizumab	ORR: 21% DCR: 37% mPFS: 10.6 weeks mOS: 30.6 weeks
**ENCORE 602/TRIO025** [24]	II	mTNBC Second or later lines (second in 69%)	41	Atezolizumab	ORR: 2.0% mPFS: 1.51 mo mOS: 12.4 mo
**TONIC** [25]	II	mTNBC First to fourth lines (first in 24%)	67	Nivolumab with (1) no induction or (2) irradiation or (3) cyclophosphamide or (4) cisplatin or (5) doxorubicin	ORR (overall cohort): 20% ORR (cisplatin cohor): 23% ORR (doxorubicin cohort): 35% ORR (TNBC cohort): 5%
**SAFIR02-BREAST IMMUNO** [26]	II	mBC First or second lines (second in 52%)	82	Durvalumab vs. chemotherapy	HR for OS (ITT): 0.84, 95 Cl: 0.54–1.29; *p* = 0.423. HR for OS (TNBC cohort): 0.54, 95% CI 0.30–0.97, *p* = 0.0377 HR for OS (TNBC PD-L1 positive) 0.37, 95% CI 0.12–1.13
**KEYNOTE-119** [19]	III	mTNBC Second or third lines (second in 60%)	622	Pembrolizumab vs. chemotherapy	mOS (PD-L1 CPS > 1): 10.7 vs. 10.2 mo (HR 0.86, 95%CI 0.69–1.06) mOS (PD-L1 CPS > 10): 12.7 vs. 11.6 mo (HR 0.78, 95%CI 0.57–1.06) OS (ITT): 9.9 vs. 10.8 mo (HR 0.97, 95%CI 0.82–1.15) PFS (ITT): 2.1 vs. 3.3 mo (HR 1.60, 95%CI 1.33–1.92)

Abbreviations: CI: Confidence interval; CPS: Combined positive score; HR: Hazard ratio; HTMB: High tumor mutational burden; ITT: Intention to treat; mBC: Metastatic breast cancer; mo: Months; mOS: Median overall survival; mPFS: Median progression free survival; mTNBC: Metastatic triple negative breast cancer; ORR: Objective response rate; OS: Overall survival; PD-L1: Programmed death ligand 1; PFS: Progression free survival; TNBC: Triple negative breast cancer.

#### 2.1.2. Immune-Checkpoint Inhibitors in Combination with Chemotherapy

As almost all previously reported studies suggested that ICIs as monotherapy might offer only limited survival benefit, subsequent studies focused on combining ICIs with chemotherapy to enhance immune response. The rationale behind this combination is to release antigens to stimulate immunogenicity and effector T-cells, as well as to reduce regulatory T-cells, increase MHC1 and IFN1 and make the microenvironment more favorable [27].

Up to now, three randomized phase III trials investigated the role of immunotherapy plus chemotherapy versus chemotherapy alone in first-line metastatic TNBC (Table 2).

The first phase III trial testing immunotherapy combined with chemotherapy versus chemotherapy alone was the IMpassion130 [28]. Patients were randomized to receive atezolizumab plus nab-paclitaxel versus nab-paclitaxel plus placebo. The two co-primary endpoints were PFS and OS. The study had a hierarchical statistical design; a significant improvement in PFS and then OS in the intent-to-treat (ITT) population was required to evaluate the same endpoint in the PD-L1 positive subgroup. PD-L1 positivity was defined by ≥1% PD-L1 expression on tumor-infiltrating immune cells. In the ITT population, the study showed a modest but significant benefit in favour of the combined arm in PFS (7.2 vs. 5.5 months, HR 0.80, *p* = 0.002), but not in OS (21.0 vs. 18.7 months, HR 0.87, *p* = 0.08). Considering only patients with PD-L1 positive disease (396, 40.9%), the addition of atezolizumab was statistically significant in terms of PFS (7.5 months vs. 5.3 months, HR 0.63, *p* < 0.001); however, no statistical benefit was found in terms of median OS (25.4 vs. 17.9 months, HR 0.67, 95%CI 0.53–0.86) [29,30].

Based on the primary results in the PD-L1 positive subgroup, atezolizumab received accelerated approval from the Food and Drug Administration (FDA) in March 2019 and from European Medicines Agency (EMA) in August 2019, making it the first immunotherapeutic agent to be approved in this setting.

Due to different accessibility in the use of nab-paclitaxel as first-line therapy worldwide, the IMpassion131 trial tested the combination of atezolizumab and paclitaxel in the same setting of patients. Differently from IMpassion130, the primary endpoint of the study was investigator-based PFS tested in the PD-L1 positive population and, if significant, in the ITT. OS was a secondary endpoint to be tested only if the PFS was positive. Surprisingly, the study did not demonstrate improved PFS in the PD-L1 positive subgroup (6.0 vs. 5.7 months, HR 0.82, *p* = 0.20) [31].

Several hypotheses have been proposed to explain these discordant results; among these, the fact that paclitaxel requires premedication with steroids to prevent hypersensitivity reactions and the immunosuppressive effect of steroids prior to immunotherapy administration could have impaired immunotherapy activity [32]. However, the extent to which steroid premedication attenuates the response to immunotherapy is only speculative and has never been confirmed so far. In addition, patients treated with chemotherapy alone in the IMpassion131 trial showed a much higher ORR and median OS than would be expected in patients with advanced TNBC. OS in the PD-L1 positive control arm of IMpassion131 was 28.3 months compared to the 17.9 months reported in IMpassion130. It has also been suggested that inherent differences in the biology of TNBC patients enrolled in the two studies, such as an imbalance between luminal (more chemotherapy-sensitive) and basal-like (chemotherapy-resistant) intrinsic subtypes, may explain these different results. The cohorts of patients in the two trials were similar in terms of age, performance status, disease setting, metastatic sites, PD-L1 expression, prior adjuvant therapy and proportion of de novo metastatic breast cancer. However, in the IMpassion131 only 13% of patients in the placebo arm of the PD-L1 positive group had more than three metastatic sites while in IMpassion130, 23.5% of patients in the chemotherapy alone arm and 19.5% of those in the atezolizumab arm had a higher burden of disease with four or more metastatic sites. In conclusion, despite several hypotheses tried to explain these contradictory results, none of them so far has been able to provide any solid conclusion.

In August 2021, Roche voluntarily withdraw the FDA accelerated approval for the combination of atezolizumab and nab-paclitaxel as a first-line treatment for metastatic TNBC in patients with PD-L1 positivity. The main reason for this choice was the approval of pembrolizumab, which brought significant changes to TNBC treatment landscape. Consequently, atezolizumab no longer met the criteria for accelerated approval. In Europe, the EMA thoroughly reviewed the results of the IMpassion131 study and concluded that atezolizumab should only be administered in combination with nab-paclitaxel.

The ongoing IMpassion132 trial, enrolling TNBC patients with early relapses (<12 months), may help clarify some doubts. In the experimental arm, atezolizumab is combined with different chemotherapy partners such as carboplatin and gemcitabine or capecitabine. This could help us clarify whether the main reason for the negative results of the IMpassion131 is atezolizumab itself or the chemotherapy backbone [33].

Besides the contradictory results with atezolizumab, pembrolizumab demonstrated more consistent data. The KEYNOTE-355 [34,35] is a phase III trial testing pembrolizumab in combination with chemotherapy of physician’s choice that could be either paclitaxel, nab-paclitaxel or gemcitabine plus carboplatin in the first-line setting. PFS and OS were the two co-primary endpoints and were assessed with a hierarchical design in three different groups: PD-L1 CPS ≥ 10, CPS > 1, and ITT population. In patients with PD-L1 expression ≥10%, as determined by CPS, which accounted for 38% of patients in the trial, PFS improved to 9.7 months with pembrolizumab plus chemotherapy compared to 5.6 months in the placebo group (HR 0.66, *p* = 0.0012), regardless of the chemotherapy partner. No statistical differences were observed in the PD-L1 CPS > 1 and in the ITT, which was not formally tested. This result led to the approval of the drug by the FDA in November 2020 for patients with PD-L1 positive CPS > 10 tumor. Final OS results with longer follow-up demonstrated a 7-month improvement with pembrolizumab, from 16.1 months to 23.0 months (HR 0.73, *p* = 0.0093). Subgroup analysis showed no benefit for patients with CPS < 10% and minimal incremental benefit for patients with tumor proportion score <20 compared with tumor proportion scores ≥10 [36]. Since approximately 30% to 40% of patients with metastatic TNBC will have CPS score ≥10 for which pembrolizumab is FDA-approved in combination with chemotherapy, all patients with advanced disease should undergo PD-L1 testing at diagnosis of advanced disease to optimize treatment decisions (Table 2).

**Table 2 cancers-15-02933-t002:** Main phase III studies with immune-checkpoint inhibitors in the first line for advanced TNBC.

Study Name		IMpassion130 [28,29,30]	IMpassion131 [31]	KEYNOTE-355 [34,35,36]
**Population**		mTNBC (=902)	mTNBC (=651)	mTNBC (=847)
**Random**		1:1	2:1	2:1
**ICI**		Atezolizumab	Atezolizumab	Pembrolizumab
**Chemotherapy**		Nab-paclitaxel	Paclitaxel	Nab-paclitaxel Paclitaxel Carboplatin and gemcitabine
**Primary endpoint**		PFS and OS in ITT and PD-L1 positive (hierarchical)	PFS in PD-L1 positive and ITT (hierarchical)	PFS and OS in PD-L1 CPS score ≥10, ≥1, and ITT (hierarchical)
**PD-L1 definition**		IC > 1	IC > 1	CPS > 1 and CPS > 10
**Assay**		SP142	SP142	22C3
**Findings**	**PFS**	ITT: 7.2 vs. 5.5 mo (HR 0.80, 95% CI 0.69–0.92, *p* = 0.002) PD-L1: 7.5 vs. 5.3 mo (HR 0.63, 95% CI 0.50–0.80)	ITT: 5.7 vs. 5.6 mo (HR 0.86, 95% CI 0.70–1.05) PD-L1: 6.0 vs. 5.7 mo (HR 0.82, 95% CI 0.60–1.12; *p* = 0.20)	ITT: 7.5 mos vs. 5.6 (HR 0.82, 95%CI 0.70–0.98) PD-L1 CPS ≥ 10: 9.7 vs. 5.6 (HR 0.66, 95%CI 0.50–0.88) PD-L1 CPS ≥ 1: 7.5 vs. 5.6 (HR 0.75, 95%CI 0.62–0.91)
	**OS**	ITT: 21.0 vs. 18.7 mo (HR 0.87, 95% CI 0.75–1.02, *p* = 0.08) PD-L1: 25.4 vs. 17.9 mo (HR 0.67, 95%CI 0.53–0.86)	ITT: 19.2 vs. 22.8 mo (HR 1.12, 95% CI 0.88–1.43) PD-L1: 22.1 vs. 28.3 mo (HR 1.11, 95% CI 0.76–1.64)	ITT: 17.2 mos vs. 15.5 (HR 0.89, 95%CI 0.76–1.05) PD-L1 CPS ≥ 10: 23.0 vs. 16.1 (HR 0.73, 95%CI 0.55–0.95, *p* = 0.0093) PD-L1 CPS ≥ 1: 17.6 vs. 16.0 (HR 0.86, 95%CI 0.72–1.04, *p* = 9.0563)

Abbreviations: CI: Confidence interval; CPS: Combined positive score; HR: Hazard ratio; IC: Immune cell score; ICI: Immune checkpoint inhibitor: ITT: Intention to treat; mo: Months; mTNBC: Metastatic triple negative breast cancer; OS: Overall survival; PD-L1: Programmed death ligand 1; PFS: Progression free survival.

### 2.2. Early Triple-Negative Breast Cancer

While the initial discoveries came from studies conducted in the metastatic setting, in general, early-stage disease represents a more promising scenario for immunotherapy for several reasons. Most importantly, the tumor burden is more limited, the disease is therefore more homogeneous from a biological point of view, and the tumor microenvironment is less immunosuppressive and less impacted by previous systemic treatments [37]. Moreover, in breast cancer, the higher rates of TILs and PD-L1 expression in the early compared to advanced stage suggest a potentially greater benefit of immunotherapy in this subset [38].

Table 3 reports the main trials evaluating immunotherapy in early TNBC. In KEYNOTE-522 [39] stage II and III TNBC patients received neoadjuvant chemotherapy (NACT) associated with concomitant pembrolizumab or placebo that continued in the adjuvant setting. The chemotherapy backbone consisted of weekly paclitaxel plus carboplatin followed by triweekly anthracycline plus cyclophosphamide. The two co-primary endpoints were pathological complete response (pCR) and event-free survival (EFS). In the first interim analysis, the study showed an increased pCR rate with pembrolizumab compared to placebo (64.8% vs. 51.2%, *p* < 0.001) with a delta of 13.6%, [39] which, interestingly, decreased to 7.5% in the third interim analysis (63% vs. 55.6%). This result may suggest that pCR is not a good surrogate for EFS results in immunotherapy trials. Indeed, the last update of the study, with a median follow-up of 39 months, confirmed the EFS benefit of ICI, as the three-year EFS rate was 84.5% with pembrolizumab versus 76.8% with placebo (HR 0.63, *p* = 0.0003) [40,41]. The final analysis also showed that pembrolizumab benefit was maintained across all subgroups and regardless of nodal status [40]. Furthermore, patients who achieved pCR demonstrated higher 3-year EFS rates, with 94.4% and 92.5% in the pembrolizumab and placebo groups, respectively. In contrast, patients who did not achieve pCR had lower 3-year EFS rates, with 67.4% and 56.8% in the pembrolizumab and placebo groups, respectively. As a result, the additional benefit derived from incorporating pembrolizumab was relatively greater in patients who did not achieve pCR (Δ = 10.6%) compared to those who achieved pCR (Δ = 1.9%). Regarding the PD-L1 status, more than 80% of patients were positive as defined by CPS > 1%, and these patients experienced higher rates of pCR (68.9% vs. 54.4%) compared to the negative cohort (45.3% vs. 30.3%). Of note, both groups benefit and thus the decision to add pembrolizumab in the neoadjuvant setting for early TNBC is independent of the expression of PD-L1 positivity [40,41]. Based on these data, on July 2021 FDA approved pembrolizumab in combination with chemotherapy for the neoadjuvant treatment of high-risk early TNBC. Specifically, it is indicated for patients with TNBC who have tumors larger than 1 cm but not exceeding 2 cm in diameter with nodal involvement, or tumors larger than 2 cm in diameter, regardless of nodal involvement [42].

The IMpassion031 trial [43] tested the combination of atezolizumab with nab-paclitaxel followed by dose-dense anthracycline-based chemotherapy. Similar to the KEYNOTE-522 trial, a higher rate of pCR was observed with the addition of immunotherapy (57.6% vs. 41.1%, *p* = 0.0044). In contrast, the PD-L1 positive cohort achieved a higher rate of pCR compared with the PD-L1 negative cohort (68.8% vs. 49.3%) but this benefit did not cross the prespecified significance boundary and thus the study was not formally powered for testing EFS, disease-free survival (DFS) and OS analyses.

The other phase III trial testing atezolizumab in the neoadjuvant setting is the NeoTRIP [44] which used a non-anthracycline backbone with carboplatin and nab-paclitaxel in the neoadjuvant setting, followed by adjuvant anthracycline-based chemotherapy. The trial failed to show an improvement in pCR with the addition of atezolizumab (48.6% vs. 44.4%, OR 1.18, *p* = 0.48). However, PD-L1 status was the greatest predictor of pCR regardless of the treatment received. High levels of PD-L1 were associated with the highest pCR rate. Stromal and intratumoral TILs were also predictive of pCR in both treatment arms. Continuing follow-up for the EFS is ongoing [45].

Another promising ICI tested in the early setting is durvalumab, which was evaluated in association with nab-paclitaxel followed by dose-dense epirubicin/cyclophosphamide in the phase II GeparNuevo trial [46]. The study failed to demonstrate statistically significant increase in pCR rates with the addition of durvalumab (53.4% vs. 44.2%, *p* = 0.287). However, a significantly improved 3-year iDFS (84.9% vs. 76.9%, HR 0.54, *p* = 0.0559) and OS (95.1% vs. 83.1%, HR 0.26, *p* = 0.0076) were reported, questioning the validity of pCR as a valid surrogate endpoint in the neoadjuvant immunotherapy trials [47]. Interestingly, pCR was only improved in patients treated in the window-of-opportunity part, where durvalumab was administered for 2 weeks prior to the start of chemotherapy.

Other ongoing trials are evaluating the role of neoadjuvant with or without adjuvant immunotherapy in early-stage TNBC. The use of atezolizumab is currently being investigated in large trials, including the neoadjuvant to adjuvant NSABP-B59/GeparDouze trial (NCT03281954) and the adjuvant IMpassion030/ALEXANDRA trials [48]. Trials assessing the efficacy of other ICIs, such as avelumab (A-BRAVE trial, NCT02926196), are also ongoing. Results from these studies will provide more useful information on the role of ICIs in early-stage TNBC as well as the best treatment schedule.

**Table 3 cancers-15-02933-t003:** Main studies with immune-checkpoint inhibitors in early TNBC.

ICI	Study Name	Phase	Population	Treatment	Findings		pCR Improvement	EFS Benefit
					pCR	EFS		
**Pembrolizumab**	**I-SPY2** [47]	II	Stage II/III HER2-negative (=250)	A: paclitaxel + pembrolizumab → AC B: paclitaxel → AC	HER2-negative: 44 vs. 17% TNBC: 60 vs. 22%	Numerically higher but not powered for statistical significance	Yes	Not reported
	**KEYNOTE-522** [39,40,41]	III	T1c N1-2 or T2-4 N0-2 TNBC (=1174)	A: carboplatin and paclitaxel + pembrolizumab → AC/EC → surgery → pembrolizumab B: carboplatin and paclitaxel + placebo → AC/EC → surgery → placebo	ITT: 63% vs. 55.6% PD-L1 positive: 54.9 vs. 68.9% PD-L1 negative: 30.3 vs. 45.4%	3-year EFS: 84.5 vs. 76.8% (HR 0.63, 95% CI 0.48–0.88, *p* = 0.0003)	Yes	Yes
**Atezolizumab**	**NeoTRIPaPDL1** [44]	III	Early high-risk or locally advanced TNBC (=280)	A: carboplatin and nab-paclitaxel + atezolizumab → surgery → AC/EC B: carboplatin and nab-paclitaxel → surgery → AC/EC	ITT: 48.6 vs. 44.4% PD-L1 positive: 51.9 vs. 48%	Not reported	Not significant	Not reported
	**IMpassion031** [43]	III	cT2-4 cN0-3 TNBC (=333)	A: Atezolizumab → nabpaclitaxel → AC B: placebo → nabpaclitaxel → AC	ITT: 57.6 vs. 41.1% PD-L1 positive: 68.8% vs. 49.3%	Not reported	Yes	Not reported
**Durvalumab**	**GeparNUEVO** [46]	II	T1b-T4a-d TNBC (=174)	A: durvalumab + nabpaclitaxel → AC B: placebo + nabpaclitaxel → AC	ITT: 53.4 vs. 44.2% PD-L1 positive: 58 vs. 50.7% window-cohort: 61.0 vs. 41.4%	3-year iDFS: 84.9 vs. 76.9% 3-year OS: 95.1 vs. 83.1%	Not significant	Yes

Abbreviations: AC: Doxorubicin and cyclophosphamide; CI: Confidence interval; CPS: Combined positive score; CT: Chemotherapy; DDFS: Distant disease free survival; EC: Epirubicin cyclophosphamide; EFS: Event free survival; HR: Hazard ratio; ICI: Immune checkpoint inhibitor; iDFS: Invasive disease free survival; ITT: Intention to treat; OR: Odd ratio; OS: Overall survival; pCR: Pathological complete response; PD-L1: Programmed death ligand 1; TNBC: Triple negative breast cancer.

## 3. Novel Strategies and Future Directions

### 3.1. Dual Immunotherapy

The phase II NIMBUS study investigated the combination of nivolumab and ipilimumab in patients with hypermutated HER2-negative breast cancer. The ORR was 16.7% for patients with TMB ≥9 mut/Mb, which could reach 60% for those with TMB ≥14 mut/Mb. Durable responses of more than one year were observed with this strategy [49]. Recently, the combination of ipilimumab and nivolumab was evaluated in the DART trial. The study met its primary endpoint with 18% ORR among advanced, chemotherapy-refractory metastatic TNBC and no new safety signals were reported [50].

The BELLINI trial investigated the use of neoadjuvant nivolumab, with or without low-dose ipilimumab, in stage I-III TNBC patients with high levels of TILs. Results presented at ESMO 2022 showed that 23% of patients has partial radiological response (PR) after only 4 weeks of neoadjuvant nivolumab treatment. Immune activation was observed in 58% of patients and all patients with PR had TIL levels above 40%. These results suggest the potential for using ICIs without chemotherapy for early TNBC patients [51].

The combination of tremelimumab, an anti-CTLA4, durvalumab and chemotherapy is being investigated in the phase I/II MOVIE trial, with moderate activity and a consistent safety profile [52] (Table 4).

### 3.2. PARPi

Poly (ADP-ribose) polymerase (PARP) inhibitors have demonstrated therapeutic efficacy in patients with germline BRCA1 or BRCA2 (gBRCA) mutations [53,54,55]. Preclinical evidence suggests that PARP inhibitors may upregulate cancer cell immunogenicity and enhance the sensitivity to immunotherapies by creating neoantigens through DNA damage, upregulating interferon production, and increasing PD-L1 expression [25]. For this reason, the combination of ICIs with PARP inhibitors is being investigated in BRCA-deficient breast cancer.

MEDIOLA was a phase II trial testing the combination of durvalumab and Olaparib in patients with germline BRCA1 or BRCA2 mutations, reporting interesting results in terms of responses (ORR of 63.3%) [56]. The combination of pembrolizumab and niraparib in the same setting of patients was tested in the TOPACIO/KEYNOTE-162 trial. The combination showed a good ORR of 21%, which was higher in patients with germline or somatic BRCA mutations (ORR 47%) [57] (Table 4).

In the randomized, non-comparative, phase II DORA trial, patients with TNBC were randomised to olaparib or olaparib plus durvalumab after a period of clinical benefit from platinum chemotherapy. The aim of the study was to test whether platinum sensitivity could serve as a sort of surrogate biomarker for patients with TNBC who might benefit from a maintenance strategy with PARP inhibitors with or without immunotherapy. The study showed that a subgroup of non-gBRCA-altered TNBC patients achieved durable DCR with a chemotherapy-free maintenance strategy of olaparib with or without durvalumab [58]. The ongoing KEYLYNK trial (NCT04191135) is testing the efficacy of pembrolizumab in combination with olaparib or chemotherapy after induction treatment with chemotherapy and pembrolizumab.

### 3.3. Angiogenesis Inhibitors 

Anti-angiogenic drugs have been investigated in combination with immunotherapy due to their immunomodulatory properties. They are able to increase lymphocytic infiltration into the tumor and enhance antitumor immune responses [59]. The FUTURE-C-PLUS trial was a phase II study aimed to evaluate the effectiveness and safety of combining the anti-PD-1 antibody camrelizumab with nab-paclitaxel and the multityrosine kinase inhibitor famitinib, targeting VEGFR-2, PDGFR, and c-kit, in patients with metastatic TNBC [60] showing good results in terms of ORR, 81% in the ITT population. Similarly, a phase II trial explored the combination of the anti-PD-1 antibody nivolumab, in combination with the anti-VEGF antibody bevacizumab and paclitaxel chemotherapy as a first-line treatment for HER2-negative metastatic breast cancer patients. This combination demonstrated highly promising results, with an ORR of 70% and a DCR of 98%. Consequently, further investigation is warranted, particularly in the HER2-negative population, including TNBC patients who were underrepresented in the study (only 18 patients, accounting for 32%) [61].

### 3.4. ADCs

Antibody-drug conjugates (ADCs) represent a group of new drugs with an innovative mechanism of action. ADCs consist of a monoclonal antibody specific for proteins overexpressed on tumor cells, linked to a cytotoxic drug or payload through a linker. 

Metastatic breast cancer often exhibits elevated levels of Trop-2, a transmembrane calcium signal transducer, which contributes to tumor growth and advancement, providing the rationale for testing anti-Trop2 treatment strategies in these patients. Sacituzumab Govitecan (SG) is a novel ADC that combines an anti-Trop2 monoclonal antibody with SN-38, the active form of irinotecan, through a specialized hydrolyzable linker. This linker facilitates the intracellular and tumor microenvironment release of SN-38 [62].

SG is approved for the treatment of metastatic TNBC based on the results of the phase III ASCENT trial, which showed both improved PFS and OS compared to the treatment of physician choice in heavily pre-treated patients [63]. Studies are currently underway testing the association of SG with pembrolizumab and atezolizumab in metastatic TNBC. SG is also being tested in the early setting, including in combination with immunotherapy in the ASPRIA trial (NCT04434040), where adjuvant SG is administered together with atezolizumab in TNBC patients with residual disease after NACT and detectable circulating tumor DNA (ctDNA). The combination of SG with pembrolizumab in patients with advanced TNBC is under investigation in the ASCENT-04 trial (NCT05382286).

The HER2-targeted ADC, trastuzumab deruxtecan (T-DXd), is approved for HER2-low metastatic breast cancer due to the improvement in PFS and OS [64]. Preclinical data have shown that the combination of ADCs with ICIs can increase the activity of immunotherapy due to direct activation of dendritic cells, increased expression of PD-L1 and enhanced formation of neoantigens [65,66].

Prior studies have yielded conflicting results regarding the prognostic value of low HER2 expression, lacking robust evidence for its independent prognostic significance. Nevertheless, recent advances in both basic and clinical research have significantly transformed our understanding of HER2-low breast cancer [67]. First, the development of innovative anti-HER2 ADCs has paved the way for ongoing trials with the potential to revolutionize the clinical management of HER2-low breast cancer. Second, there is growing recognition of HER2-low breast cancer as a distinct clinical and biological entity, potentially influencing patient prognosis. Consequently, accurate identification and classification of patients with HER2-low breast cancer, including those initially diagnosed with TNBC, is essential as it enables the implementation of novel treatment strategies specifically tailored for this subgroup.

BEGONIA (NCT03742102) is an ongoing 2-part open-label platform study evaluating in first-line TNBC the safety and efficacy of durvalumab in combination with other novel therapies, such as the two novels ADC, dapotomab deruxtecan (Dato-DXd) and T-DXd. In the most recent updated analysis, the combination of durvalumab with Dato-DXd showed compelling high response rates (ORR 74%) with promising durability, irrespective of PD-L1 expression [68]. Similarly, combining durvalumab with T-DXd in HER2-low metastatic breast cancer yielded a promising ORR of 57% along with encouraging PFS data and durable responses [69]. Both studies reported manageable adverse events consistent with the known safety profiles of each agent [68,69].

### 3.5. Current and Future Directions

New strategies are currently under investigation. The IMPRIME-1 trial tested the efficacy of Imprime-PGG, an innate-immune activator, in combination with pembrolizumab for heavily pretreated metastatic TNBC patients. The trial showed an ORR of 15.9%, a DCR of 54.5%, and a median OS of 16.4 months, compared to 9 months for those receiving pembrolizumab alone [70].

Another approach is to use purinergic pathway antagonists such as oleclumab, which was tested in a phase I/II SYNERGY trial. Results presented at ESMO 2022 showed that the addition of oleclumab to durvalumab with carboplatin/paclitaxel did not increase clinical benefit rate at week 24 in first-line advanced TNBC, although 9 long responder patients in both treatment arms are still under immunotherapy maintenance [71]. Other combination strategies being tested include targeting the MAPK pathway with cobimetinib, using CDK inhibitors such as dinaciclib and palbociclib, and using histone deacetylase inhibitors such as romidepsin and entinostat. Translational analyses are ongoing to better understand the heterogeneity of TNBC and improve treatment outcomes [12].

Early-phase trials are also investigating the efficacy and safety of dendritic cell (DC) based antitumor vaccines. In HER2-negative patients, the addition of DC vaccines to standard NACT improved the rate of pCR from 2.8% to 23.1% in the PD-L1-negative population, although no significant differences were observed in 7-year EFS or in OS [72]. CAR-T cell therapy, a form of adoptive cell therapy in which autologous T cells are re-engineered to bind cancer antigens, and bispecific T-cell engagers, are under investigation for the treatment of breast cancer.

## 4. Open Question about the Use of Immunotherapy in Triple-Negative Breast Cancer

Some important issues regarding the use of immunotherapy in clinical practice exist and need further investigation. These open questions are shown in Figure 1 and include (not-extensive list): (I) the validity of PD-L1 as a biomarker, (II) the need to identify other biomarkers predictive of response, (III) the question of the best chemotherapy backbone, (IV) how to define the duration of immunotherapy, (V) what to do in patients without pCR after neoadjuvant immunotherapy, (VI) what to do in patients with early relapse after (neo)adjuvant immunotherapy, (VII) the need for different endpoints and criteria to assess response to immunotherapy, (VIII) efforts to minimize the impact of immune-related adverse events (irAEs), and (IX) financial toxicities.

### 4.1. PD-L1 as Valid Biomarkers

While PD-L1 proved to be a predictive factor of immunotherapy in several studies in the metastatic setting, its predictive role was not confirmed in other trials.

In addition, the development of different immunohistochemical assays, scoring systems, and cutoff values to determine PD-L1 status in the IMpassion130 and KEYNOTE-355 trials generates confusion about how to assess PD-L1 status in routine clinical practice; the various assays and scoring systems are not interchangeable and it is important that clinicians and pathologists are aware of this. The status of PD-L1 was defined differently among the studies: in the IMpassion trials, PD-L1 was considered positive if the immune cell score was 1% or greater, using the SP142 assay (Ventana; Roche, Basel Switzerland). In the KEYNOTE-355, the PD-L1 positivity was defined by the CPS using the 22C3 assay (Dako/Agilent Technologies, Santa Clara, CA, USA), with CPS cutoffs of 1% or greater and 10% or more. Despite attempts to harmonize scoring systems, agreement between tests remains low. Each test may miss about 20% of patients who are classified as positive by another test. It may not be practical for laboratories to perform multiple assays, and this problem has yet to be resolved.

Although the expression of PD-L1 seems an established predictive biomarker in the advanced setting, it did not differentiate between responders and non-responders in the early setting. However, it is important to note that the PD-L1 threshold adopted in the subgroup analyses of KEYNOTE-522 (CPS ≥ 1) may not be the optimal one, since a threshold of CPS ≥ 10 is currently used to select patients in the metastatic setting, warranting this additional analysis in the future.

To make things worse, the temporal and spatial heterogeneity of PD-L1 should be taken into account. The levels of PD-L1 expression can vary between different stages of the disease, with higher levels observed in primary tumors than in metastatic lesions. Moreover, the expression levels may differ depending on the location of the metastatic site, with lung, soft tissues and lymph nodes metastases showing higher expression levels than liver, skin and bone metastases [73]. In the IMpassion130, a lower average PD-L1 positivity was reported in metastatic biopsies compared with primary tumors (36% vs. 44%, *p* = 0.014), and among the metastatic sites, PD-L1 expression was lowest in the liver and highest in the lymph nodes [74]. These discrepancies highlight the importance of considering the temporal and spatial heterogeneity of PD-L1 expression in breast cancer when interpreting PD-L1 test results. Clinicians need to be aware of these variations and carefully consider the interpretation of PD-L1 results in each individual patient.

### 4.2. Other Predictive Biomarkers 

In clinical trials the response to ICIs is very heterogeneous and discovering new predictive biomarkers is un unmet need. However, to date, none of the translational research efforts have identified clinically useful biomarkers that can accurately predict the response to immunotherapy [75].

Biomarkers currently under investigation include TILs, immune phenotype, tumor mutational burden, and BRCA1/2.

It has been suggested that a high number of TILs in the pembrolizumab arm of the KEYNOTE-119 trial correlate favorably with clinical outcomes [76]. In a biomarker analysis of IMpassion130 trial data, Emens et al. showed that CD8-positive intratumoral TILs and stromal TILs were associated with improved outcomes with atezolizumab [77]. Furthermore, TILs have been shown to have strong prognostic value for early-stage TNBC. Of note, in the GeparNuevo trial, the presence of TILs seemed to predict benefit in both the durvalumab-containing arm and the placebo arm, calling into question the use of this biomarker alone to select patients for immunotherapy [78]. 

Immune phenotype refers to the immune cell composition and activation status of the tumor microenvironment, which can be assessed by gene expression profiling or immunohistochemistry. There is a plethora of biomarkers that identify “immune-enriched” tumors and are associated with a greater magnitude of benefit within the PD-L1 positive population. Immune-enriched profile is related to a basal-like immune-activated molecular subtype, compared with a basal-like immune-suppressed subtype. 

TMB reflects the number of somatic mutations in the tumor genome, which can be measured by next-generation sequencing. About 5% of metastatic breast cancer is associated with high TMB and has been shown to correlate with good response to immunotherapy [79]. For these patients, pembrolizumab monotherapy has a tumor-agnostic approval based on the results of the KEYNOTE-158 trial [21].

The presence of BRCA 1/2 mutations provides higher genomic instability, leading to increased neoantigen load, making these tumors a very attractive target for immunotherapy. However, Emens et al. demonstrated that the presence of a germline BRCA1/2 mutation was not associated with PD-L1 expression and did not predict the benefit from atezolizumab in the IMpassion130 trial [77]. Current studies are focusing on identifying immune cell signatures that are able to better identify patients with immunotherapy-responsive disease. Some promising examples are a 27-gene TME assay [80], enhanced immune (Immune+) [81] and the ImPRINT assay [82].

### 4.3. Backbone Chemotherapy

The low efficacy of immunotherapy as a monotherapy in TNBC, as well as the increased response rates seen with combination regimens comprising immunotherapy and chemotherapy, suggest that chemotherapy may influence the tumor microenvironment and the likelihood of response to ICI based regimens [25]. 

Additionally, preliminary clinical findings indicate that the type of chemotherapy employed could influence both the tumor microenvironment and the probability of a response to immunotherapy-based regimens. For instance, in the TONIC trial, metastatic TNBC patients received a 2-week low-dose induction with various chemotherapy agents, including cyclophosphamide, cisplatin, doxorubicin, or irradiation, all followed by nivolumab. The cisplatin and doxorubicin cohorts demonstrated the highest ORR and also displayed an upregulation of immune-related genes implicated in PD-1/PD-L1 and T-cell cytotoxicity pathways [25].

The NeoTRIP trial, which did not include anthracyclines in the neoadjuvant setting, failed to demonstrate an increased pCR. This contrasts with the positive result of IMpassion031 and thus suggests that anthracycline induction may lead to an upregulation of immune-related genes, thereby priming the tumor microenvironment for a more favorable response to immunotherapy [44].

### 4.4. Optimal Duration of Immunotherapy

There is ongoing debate about how long immunotherapy treatment should be administered. Large, randomized trials have shown that immunotherapy can produce long-lasting responses, raising questions about whether it should be continued until disease progression or unacceptable toxicity in the metastatic setting. 

The appropriate duration of immunotherapy in the early setting is even more uncertain. Most trials do not differentiate the contributions of the neoadjuvant and adjuvant treatment phases, therefore it is not clear whether immunotherapy should be administered only in the neoadjuvant setting or also post-surgery and if so, for how long. While immunotherapy was continued after surgery in IMpassion031 and KEYNOTE-522 trials, GeparNuevo did not continue durvalumab postoperatively and still showed improvement EFS, suggesting that most of the benefit of ICI therapy is achieved in the neoadjuvant phase. Restricting treatment to the neoadjuvant phase could be associated with important benefits such as reducing treatment burden and lowering costs. 

Biomarkers are not yet able to accurately identify which patients will benefit from ICIs in the early setting. Ongoing trials are evaluating different possibilities of ICI timing, but final answers will only be available after future studies designed to directly compare these different approaches.

### 4.5. Patients without pCR after Neoadjuvant Immunotherapy

The standard of care for patients with residual disease after neoadjuvant therapy was adding capecitabine according to the results of CREATEx trial [83], but whether there is any benefit of adding capecitabine after ICI therapy remains unclear. A similar dilemma exists for patients with germline BRCA1/2 mutations, for whom a role for adjuvant olaparib has recently been established in case of high-risk early TNBC and/or residual disease post neoadjuvant chemotherapy questioning the optimal positioning of ICI in this setting [53]. A suggested algorithm could be for BRCA wild-type patients to continue treatment with pembrolizumab along with capecitabine, or to stop pembrolizumab after surgery and add capecitabine soon after. For patients with BRCA mutations, there could be the option of continuing pembrolizumab along with olaparib, or continuing pembrolizumab and then adding olaparib, or stopping pembrolizumab after surgery and adding olaparib soon after. Further trials are required to address these questions. 

### 4.6. Patients with Early Relapse after (Neo)Adjuvant Immunotherapy

The use of immunotherapy in early-stage TNBC raises questions about what to do for patients who experience early relapse after (neo)adjuvant immunotherapy. It is unclear if patients can be rechallenged with immunotherapy and if disease-free interval matters. Trials in metastatic setting only enrolled immunotherapy-naïve patients and thus new clinical trials addressing this question are urgently needed. Patients with primary resistance who have a high residual cancer burden at the time of surgery, or early recurrences after surgery, are likely to have different disease biology and mechanism of resistance than patients who develop late recurrence. Whether retreatment with ICI is useful in patients with late relapse is unknown. Understanding the mechanisms of resistance is key to identifying novel treatment strategies for patients with recurrent or advanced TNBC after ICI therapy, like combining ICIs with other immunotherapeutic drugs or drugs with different mechanisms of action such as PARP-inhibitors. In the meantime, in the absence of such data, chemotherapy or new ADCs remain a possible option for patients resistant to immunotherapy. 

### 4.7. Endpoints and Criteria to Assess Response to Immunotherapy in Clinical Trials

Most trials investigating ICIs in different tumor types have demonstrated a significant survival benefit, despite occasional disappointing results in terms of response rate, raising several questions about the appropriate endpoints to evaluate response to ICIs in clinical trials. ICIs work by enhancing the immune system’s response to cancer cells, leading to distinct tumor responses that may be more long-lasting than those produced by cytotoxic chemotherapy agents. Consequently, evaluating the effectiveness of ICIs requires different endpoints than those used for chemotherapy. While chemotherapy acts directly on tumor cells, ICIs restore T-cell infiltration in the tumor microenvironment, inducing a specific immune response against cancer cells [84]. As a result, trials investigating ICIs in various types of cancer have shown a significant improvement in OS despite lower response rates. This happened in the GeparNuevo trial, where there was no pCR benefit but the final analysis showed improvement in 3-year DFS and OS. The question of whether pCR might be a good surrogate for OS in patients receiving neoadjuvant treatment has always been the subject of intense scientific debate [85].

These results call into question the optimal endpoints for assessing ICI efficacy in clinical trials. For ICIs, pCR may not be the best measure of antitumor activity, and survival endpoints such as EFS may be more appropriate. However, survival endpoints require longer follow-up periods, more events, and larger sample sizes, making trial management more complex and costly.

### 4.8. Immune-Related Adverse Event (irAEs)

irAEs are autoimmune conditions that have the potential to impact any organ in the body, distinguishing them from adverse events typically associated with chemotherapy (Figure 2). Among patients with breast cancer, the most prevalent irAEs are dermatitis (occurring in up to 49% of cases), diarrhea/colitis (affecting approximately 20% of patients), and endocrinopathies such as thyroid disorders, hypophysitis and adrenal insufficiency (with incidence rates of up to 18%, up to 10%, and 5%, respectively). Diagnosing endocrine toxicities can be challenging due to non-specific symptoms, and these conditions often necessitate lifelong replacement therapy. While rare, certain irAEs have the potential to be life-threatening, including pneumonitis (5% incidence), myocarditis (5% incidence), and pancreatitis (3% incidence).

Fatigue is also commonly observed, although usually in a mild form, with an estimated overall incidence of 16–24%. Although retreatment with ICIs can be considered for most grade 1–2 irAEs, ICIs are usually permanently discontinued for grade 3–4 events [86].

The occurrence of irAEs is a serious issue especially when treating patients in a potentially curative setting, as some of these adverse effects may be permanent or life-threatening. In the KEYNOTE-522 [39] irAEs of any grade occurred in 33.5% of patients receiving pembrolizumab compared with 11.3% in the placebo group and the most common were disthyroidism (hypothyroidism 15.1%, hyperthyroidism 5.2% and thyroiditis 2%) adrenal insufficiency in 2.6%, pneumonitis in 2.2% and hypophysitis in 1.9%. One patient in the pembrolizumab arm died of immune-mediated encephalitis and another of pulmonary embolism, which was deemed immune-related. 

Furthermore, considering that a large number of patients receiving early-stage ICI are younger, fertile women, it is imperative to also evaluate the potential impact of ICI on gynecological function. According to the evidence currently available, these compounds could potentially cause libido and sexual dysfunction as well as primary and secondary hypogonadism [87].

Due to the existence of permanent and potentially life-threatening irAEs, the risk of developing these toxicities must be weighed against the absolute EFS benefit offered by neoadjuvant immunotherapy. Compared to clinical trials, in daily practice not all patients are sufficiently fit to receive immunotherapy. 

At ESMO 2022, results for the secondary endpoints assessing patient-reported outcomes (PROs) in KEYNOTE-522 were reported [88]. The study analyzed the completion of EORTC QLQ-30 and QLQ-BR23 questionnaires during the neoadjuvant and adjuvant stages and found no significant differences in PRO scores, including global health status, quality of life, emotional functioning, physical functioning, and breast symptoms, between the treatment groups. These findings suggest that the addition of pembrolizumab to chemotherapy does not negatively impact the overall quality of life. However, only patients who benefited from treatment in the neoadjuvant phase without irAEs and who continued in the adjuvant stage of the trial completed the PRO questionnaires, while those who experienced immune-related adverse events that led to treatment discontinuation may not have been included in the PRO assessments questioning the results of such an analysis.

Moreover, appropriate reporting of irAEs in clinical trials is of paramount importance. Rare irAEs might be unrecognized if clinical trial publications report only AEs occurring above a specified incidence (e.g., >10%). This could favor underreporting of rare Aes, such as cardiac events, and could mask the real incidence of some rare toxicities [89].

### 4.9. Financial Toxicity

Another major problem is the economic challenge of using expensive drugs in the treatment of TNBC. The discovery of new strategies exacerbates inequalities in care and outcomes among cancer patients. In addition, clinical guidelines often do not take into account the context of different health systems and provide algorithms that are often incoherent with national cancer policies. Therefore, effective and integrated interventions need to be developed and implemented at multiple levels. Above all, the need to approve for clinical practice only drugs with demonstrated efficacy should be mandatory.

## 5. Conclusions

ICIs in combination with chemotherapy are approved for treatment of both early and advanced TNBC. However, there are still grey areas in the use of immunotherapy in TNBC that require a deeper investigation.

In the advanced setting, the KEYNOTE-355 has solidified the role of pembrolizumab in first-line metastatic TNBC, but some questions still remain due to the formally inconclusive data of IMpassion130 and the clinical failure of IMpassion131. Moreover, the initial faith in the predictive role of PD-L1 is now largely questioned by its dynamic heterogeneity (spatial and temporal), analytic inconsistency (different assays, scoring systems and cutoff values) and futility in early TNBC. Nonetheless, PD-L1 still remains the sole biomarker required for ICI recommendation in advanced TNBC, although it seems more able to exclude rather than select candidates for immunotherapy. Accordingly, further investigations are needed to identify useful biomarkers for chemo-immunotherapy, especially in the context of the evolving treatment scenario of metastatic TNBC. 

In the early setting, KEYNOTE-522 demonstrated significant pCR and EFS benefit for the addition of pembrolizumab to the (neo)adjuvant chemotherapy regardless of PD-L1 status. Nevertheless, the final benefit in pCR rate was not as pronounced as expected. As for EFS, it is not entirely clear the role of the anti-PD1 in the neoadjuvant versus adjuvant phase, since the trial design prevents to dissect the other contribution. In the adjuvant phase, the addition of pembrolizumab in patients who achieved pCR provided only marginal benefits, while it was more significant for those with residual disease, albeit in the context of a suboptimal control arm. As a consequence, uncertainties still exist in the adjuvant recommendation of ICI after achieving a pCR and new data from ongoing clinical trials are eagerly awaited.

Over the last few years, significant steps forward have been made in the field of immunotherapy for TNBC. Nevertheless, it is crucial to recognize that the overall value of a drug is determined by the convergence of evidence from various clinical trials, rather than isolated yet relevant findings. Therefore, further insights are needed in the areas where data are limited and uncertainties persist.

## Figures and Tables

**Figure 1 cancers-15-02933-f001:**
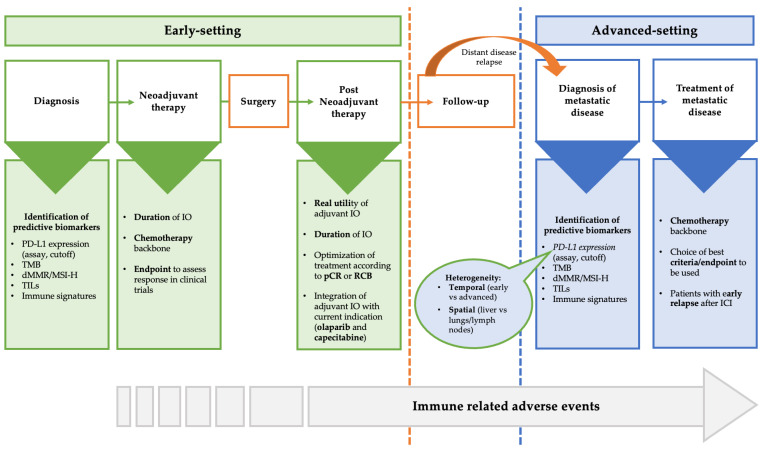
Areas of uncertainty and open question in the treatment of TNBC in the early and advanced setting. Abbreviations: dMMR: Mismatch repair deficient; ICI: Immune checkpoint inhibitors; MSI-H: Microsatellite instability-high; pCR: Pathological complete response; TILs: Tumor infiltrating lymphocytes; TMB: Tumor mutational burden; RCB: Residual cancer burden.

**Figure 2 cancers-15-02933-f002:**
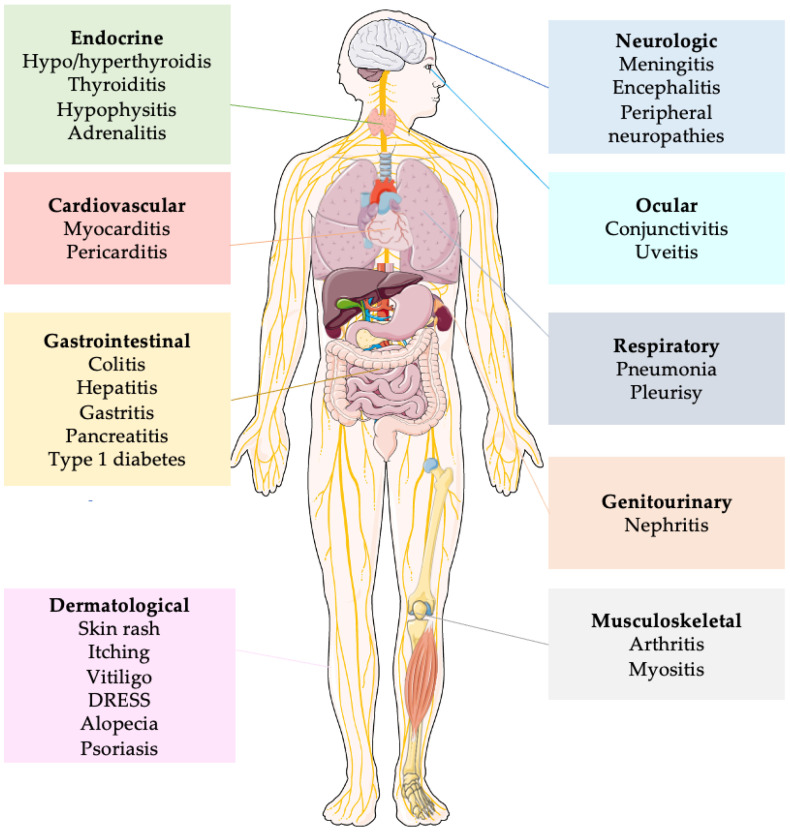
Simplified representation of principal immune related adverse events. Abbreviations: DRESS: Drug Reaction with Eosynophilia and Systemic Symptoms.

**Table 4 cancers-15-02933-t004:** Main trials of PD-(L)1 inhibitors in combination with novel strategies.

Combinations of ICIs with	Study Name	Phase	Status	Population	Treatment	Findings
**Dual immunotherapy**	**MOVIE** **(NCT03518606)**	Ib/II	Active, not rectuiting	Pretreated MBC	Durvalumab + Tremelimumab + metronomic vinorelbine	ORR 20.7%
	**NIMBUS** **(NCT03789110)**	II	Active, not rectuiting	Hypermutated HER2 negative mBC	Nivolumab + Ipilimumab	ORR (TMB ≥ 9 mut/Mb): 16.7% ORR (TMB ≥ 14 mut/Mb): 60%
	**DART** **(NCT02834013)**	II	Active, not rectuiting	Advanced MpBC	Nivolumab + Ipilimumab	ORR 18%
	**BELLINI** **(NCT03815890)**	II	Recruiting	Stage I-III TNBC	4 weeks of neoadjuvant Nivolumab +/− Ipilimumab low dose	PR: 23% Immune activation: 58%
**PARPi**	**MEDIOLA** **(NCT02734004)**	Ib/II	Active, not rectuiting	BRCA mutated HER2-negative MBC	Durvalumab + Olaparib	DCR at week 12: 80% DCR at week 28: 50% ORR: 63.3%
	**Topacio/KEYNOTE-162** **(NCT02657889)**	I/II	Completed	Pretreated mTNBC	Pembrolizumab + Niraparib	ORR (ITT): 21% ORR (PD-L1 positive): 32% ORR (PD-L1 negative): 8% ORR (gBRCAmut): 47%
	**DORA** **(NCT03167619)**	II	Completed	Platinum-treated mTNBC	Durvalumab + Olaparib vs. Olaparib	mPFS: 6.1 vs. 4.0 mo mPFS (gBRCA mut vs. wt): 8.2 vs. 2.9 mo
	**KEYLINK** **(NCT04191135)**	II	Active, not rectuiting	mTNBC after induction with first-line CT + Pembrolizumab	Pembrolizumab + Olaparib vs. Pembrolizumab + CT	Not reported
	**NCT03594396**	II	Active, not rectuiting	Preoperative treatment before NACT for StageII/III early TNBC	Durvalumab + Olaparib	pCR: 75% pCR (gBRCA): 84.6%
**Anti-angiogenic drugs**	**FUTURE-C-PLUS** **(NCT04129996)**	II	Active, not rectuiting	First-line treatment in mTNBC	Camrelizumab + nab-paclitaxel and famitinib	ORR: 81.3% mPFS: 13.6 mo mDOR: 14.9 mo
	**NEWBEAT (WJOG9917B)**	II	Active, not rectuiting	First-line treatment in HER2 negative mBC	Nivolumab + Bevacizumab and paclitaxel	ORR: 70% DCR: 98% mPFS: 14 mo mOS: 32.2
**ADCs**	**ASCENT-04(NCT05382286)**	III	Recruiting	Previously untreated PD-L1-positive mTNBC	Pembrolizumab + SG vs. pembrolizumab + CT	Not reported
	**ASPRIA** **(NCT04434040)**	II	Recruiting	Early-stage TNBC with residual disease after NACT	Atezolizumab + SG	Not reported
	**NCT04468061**	II	Recruiting	PD-L1-negative mTNBC	Pembrolizumab + SG vs. SG	Not reported
	**BEGONIA** **(NCT03742102)**	Ib/II	Active, not rectuiting	First-line treatment in mTNBC	Durvalumab + Dato-DXd	ORR: 74%
	**BEGONIA** **(NCT03742102)**	Ib/II	Active, not rectuiting	First-line treatment in mTNBC	Durvalumab + T-DXd	ORR: 57%
**Other approaches**	**IMPRIME-1(NCT02981303)**	II	Completed	Pretreated mTNBC	Pembrolizumab + Imprime-PGG	ORR: 15.9% DCR: 54.5% mOS: 16.4 vs. 9 mo
	**SYNERGY**	Ib/II	Active, not rectuiting	First-line treatment for mTNBC	Durvalumab + Oleclumab + CT vs. Durvalumab + CT	CBR: 43 vs. 44% mPFS: 6 vs. 7.7 mo

Abbreviations: CBR: Clinical benefit rate; CT: Chemotherapy; Dato-DXd: Dapotomab deruxtecan; DCR: Disease control rate; DOR: Duration of response; HER2: Human epidermial growth factor receptor 2; ITT: Intention to treat; mo: Months; mOS: Median overall survival; mPFS: Median progression free survival; mBC: Metastatic breast cancer; MpBC: Metaplastic breast cancer; mTNBC: Metastatic triple negative breast cancer; NACT: Neoadjuvant chemotherapy; ORR: Objective response rate; OS: Overall survival; PARPi: Poly ADP ribose polymerase inhibitors; PD-L1: Programmed death ligand 1; PFS: Progression free survival; SG: Sacituzumab govitecan; T-DXd: Trastuzumab deruxtecan; TNBC: Triple negative breast cancer; wt: Wild type.

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
