# Peer review of "Hope and Hype around Immunotherapy in Triple-Negative Breast Cancer"

_cancers, 2023, doi:10.3390/cancers15112933_

Round 1

Reviewer 1 Report

Excellent summary of the current standing of immunotherapy for TNBC. 

My comments:

Line 263, clarify what high risk means (tumor size, nodal involvment)

Line 379, more discussion in this section of how many TNBC patients will be able to 'shift' over to a HER2 low category for ADC based therapy. 

Section 4.1. You may as well mention specific PD-L1 antibody clones, since they are are used frequently in this type of discussion. I would also mention which PD-L1 assay is used for pembro indication.

Author Response

Thank you for your positive feedback on our manuscript; your comments have enhanced our work's overall interest and quality. Please find below our response to your comments:

- In line 263, we have addressed the clarification of the indication for high-risk early TNBC, along with the inclusion of a relevant reference.

- In line 379, we have included a concise paragraph that highlights the significance of the HER2-low classification. We emphasise that some TNBC patients may be reclassified as HER2-low, which created opportunities for them to benefit from novel treatment strategies.

- In section 4.1, we fully agree with your suggestion and have incorporated additional comments regarding the status of PD-L1. Specifically, we have mentioned the differing definitions of PD-L1 status between the IMpassion and KEYNOTE trials. Furthermore, we have provided specific information about the PD-L1 assay used for the indication of pembrolizumab.

Reviewer 2 Report

The authors have written a very good review on both hope and hype of immunotherapy for TNBC. They have summarized the main findings of clinical trials, using ICI as monotherapy, in combination with chemotherapy. They discussed novel strategies and future directions. Then they discussed in length open questions on the use of immunotherapy for this type of breast cancer. It will be useful article for novice and experts in the field alike.

I have only a few minor issues.

1.      Section 3.5. Future approaches? Those ongoing or completed may not be called “future approaches”. Maybe it is more appropriate to have a heading “current and future directions”?

2.      Current References:

(1). Please delete the list of day/month (and it is not even in English). In addition, format is not consistent from one reference to another.

(2). There are minor errors/misses in the following references:

A number of references are published abstracts: ref #34; 46; 47; 49; 50; 56; 65; 66; 68; 76; 82; 86. For some of them, complete information (e.g., source of abstract) is not provided.

Ref #34. It looks like an abstract. However, the final results on this phase III have been published by, Cortes J, et al. Lancet. 2020; 396:1817-1828 (ref #33). Therefore, the question is, does ref #34 provide additional value?

Ref #79. An abstract or an article? Please update.

Ref #82. The authors’ last names are not provided as only single letters are listed. Page numbers are missing too.  The correct information is, Agostinetto E, Gligorov J, Piccart M. Nat Rev Clin Oncol. 2022;19:763-774.

3.      Additional references may be useful. I found two recent great reviews on the related topic (this reviewer is not an co-author for these papers) and you might find useful to cite in the context of relevant discussion.

Tarantino P. et al., Immunotherapy for early triple negative breast cancer: research agenda for the next decade. NPJ Breast Cancer. 2022; 8(1):23.

Li L et al., Immunotherapy for Triple-Negative Breast Cancer: Combination Strategies to Improve Outcome. Cancers (Basel). 2023; 15(1): 321.

Author Response

We would like to express our appreciation for the thorough analysis of our manuscript and the suggestions you have provided. Please find below our response to your comments:

- We completely agree with your suggestion, and as per your recommendation, we have changed the title of the paragraph in section 3.5 from "future approaches" to "current and future directions."

- We have meticulously reviewed all the references and made the necessary adjustments as per your suggestion.

- Ref #34 has been removed

- Ref #79, which is an abstract, has been cited correctly

- Ref #82 has been reported correctly

Additionally, we have included the two additional references you have proposed.
